

# The 21st Century Decline in Damaging European Windstorms

Laura C. Dawkins[1], David B. Stephenson[1], Julia F. Lockwood[2], and Paul E. Maisey[2]

[1]College of Engineering, Mathematics and Physical Sciences, University of Exeter, Exeter, UK
[2]Met Office Hadley Centre, Exeter, UK

*Correspondence to:* Laura C. Dawkins (l.c.dawkins@exeter.ac.uk)

**Abstract.** A decline in damaging European windstorms has led to a reduction in insured losses in the 21st century. This decline is explored by identifying a damaging windstorm characteristic and investigating how and why this characteristic has changed in recent years. This novel exploration is based on 6103 high resolution model generated historical footprints (1979-2014) representing the whole European domain.

The footprint of a windstorm is defined as the maximum wind gust speed to occur at a set of spatial locations over the duration of the storm. The area of the footprint exceeding $20\,\mathrm{ms^{-1}}$ over land, $A_{20}$, is shown to be a good predictor of windstorm damage. This damaging characteristic has decreased in the 21st century, due to a statistically significant decrease in the relative frequency of windstorms exceeding $20\,\mathrm{ms^{-1}}$ in north-west Europe. This is explained by a decrease in the quantiles of the footprint wind gust speed distribution above approximately $18\,\mathrm{ms^{-1}}$ at locations in this region.

Much of the change in $A_{20}$ is explained by the North Atlantic Oscillation (NAO). The correlation between winter total $A_{20}$ and winter averaged mean sea-level pressure resembles the NAO pattern, shifted eastwards over Europe, and a strong positive relationship (correlation of 0.715) exists between winter total $A_{20}$ and winter averaged NAO. The shifted correlation pattern, however, suggests that other modes of variability may also play a role in the variation in windstorm losses.

## 1 Introduction

Weather related natural disasters cause considerable damage and economic loss globally, currently estimated to cost the insurance industry $221 billion annually (Mildenhall et al., 2014). Extratropical cyclones, also known as windstorms, are the 2nd largest cause of global insured loss (Martínez-Alvarado et al., 2014), and are a major contributor to losses in Europe. For example European windstorm Daria (24th - 26th January 1990), the most damaging event on record, incurred $8.2 billion of insured loss (indexed to 2012, Roberts et al. 2014). In addition, windstorms often arrive in quick succession, increasing the risk of large aggregate losses (Vitolo et al., 2009). Indeed, windstorms Vivian, Herta and Wiebke occurred closely after windstorm Daria, in February 1990, costing the insurance industry an additional $5.6, $1.5 and $1.4 billion respectively (indexed to 2012, Roberts et al. 2014).

However, a decline in European windstorm losses has been identified by the re/insurance industry over approximately the last two decades (Mark, 2013). This is exemplified by the winter of 2013/14, in which no individual notably damaging windstorm occurred, in spite of a strong jet stream causing an unusually large number of events to pass over Europe (Slingo et al.,



2014). The most notable events were windstorms Christian (27th - 29th October 2013) and Xaver (4th - 6th December 2013) which incurred losses of $1 and $1.5 billion respectively, much less than the most extreme windstorms in the winter of 1989/90, Daria and Vivian. This motivates the question; how and why have windstorm characteristics changed in recent decades?

Feser et al. (2015) provide a recent review of studies exploring the temporal variability of European storms. These studies used various measures of storm severity, for example surface pressure, surge height and wind speed, and give conflicting trends in storm characteristics due to differences in the data, study region and length of historical period used. For example, Donat et al. (2011) used atmospheric model generated surface pressure fields for the period 1871 to 2008 and identified a significant increasing trend in northern, central and western Europe, while Smits et al. (2005) used observed wind speed data in the Nether-
lands and found a decline in the frequency of wind speed threshold exceedance between 1962 and 2002. Cusack (2013) used a loss function, based on the exceedance of observed wind gust speeds above a damage threshold at four observation locations, to explore the change in windstorm losses in the Netherlands from 1910-2011. A multi-decadal cycle was found, with lows in the cycle in the 1960's and since 2010. Cusack (2013) identified that both lows were driven primarily by a decrease in the frequency of very damaging storms, however the present-day minima was also found to be due to a lower frequency of weaker
storms.

Windstorms are affected by large-scale weather patterns and atmospheric oscillations. One such pattern is the North Atlantic Oscillation (NAO), which describes the pressure variability between the Icelandic Low and Azores High. The NAO is the dominant mode of lower to mid-tropospheric pressure variability over the North Atlantic (Pinto et al., 2012). It therefore has a large
influence on weather conditions over the North Atlantic basin and Western Europe, and hence the occurrence and intensity of European windstorms (Pinto et al., 2012; Economou et al., 2014). Cusack (2013) speculated that a different mix of climate forcing mechanisms were influencing storms in the two identified low periods in the footprint loss function, but did not explore this relationship explicitly.

Here, the severity of a windstorm hazard event will be characterised by its footprint, defined as the maximum wind gust speed to occur at a set of spatial locations over the duration of the storm. This investigation will exploit a recent, large data set of 6103 reanalysis historical windstorm footprints over the whole European domain from October 1979 to March 2014, regionally downscaled to a 25 km grid cell horizontal resolution (see Section 2). Figure 1 shows a comparison of four such windstorm footprints, associated with the two most extreme windstorm events in each of the October - March winters, 1989/90
and 2013/14. These footprints have differing characteristics, both between and within each winter. This study aims to understand how and why footprints characteristics have changed, causing the decline in windstorm losses in recent decades. In addition, the large data set of windstorm footprints will allow for a detailed spatial exploration of this change over the whole of Europe.



Within this study a windstorm footprint characteristic that represents wind related damage will first be identified, and then used to answer two main questions:

- How has this damaging footprint characteristic changed in recent decades?

- Why has this damaging footprint characteristic changed in recent decades?

The windstorm footprint data set is introduced in Section 2. Damaging footprint characteristics are discussed in Section 3, the
above questions are then addressed in Sections 4 and 5 and future research directions are presented in Section 6.

## 2   Data

The windstorm footprint data set used in this study is the same as the raw footprint data presented in Roberts et al. (2014) with two additional extended winters (October-March) included: 2012/13 and 2013/14 (kindly provided by J. Standen and J. F. Lockwood at the Met Office).

The windstorm footprint is defined as the maximum three second wind gust speed (in ms$^{-1}$) at grid points in the region 15°W to 25°E in longitude and 35°N to 70°N in latitude over a 72 hour period centred on the time at which the maximum 925hPa wind speed occurred over land. The 925hPa wind speed is taken from ERA-interim reanalysis (Dee et al., 2011). A 72 hour duration, commonly used in the insurance industry (Haylock, 2011), is thought to capture the storms during their
most damaging phase (Roberts et al., 2014). The three second wind gust speed has been shown to have a robust relationship with storm damage (Klawa and Ulbrich, 2003), and is commonly used in catastrophe models, for risk quantification by the re/insurance industry (Roberts et al., 2014).

A total of 6301 windstorm events occurred within the domain of interest during the 35 extended winters (October - March
1979/80 - 2013/14). The storms were identified using the objective tracking approach of Hodges (1995). Extended (October - March) winters are used because they correspond to the period of the year in which most windstorms occur over central Europe (Klawa and Ulbrich, 2003).

A footprint is created for each of the 6301 events by dynamical downscaling. ERA-Interim reanalysis (Dee et al., 2011) is
downscaled to a horizontal resolution of 25km using the Met Office unified model (MetUM). The wind gust speeds are calculated from wind speeds in the MetUM model, based on a simple gust parameterisation $U_{gust} = U_{10m} + C\sigma$, where $U_{10m}$ is the wind speed at 10 metre altitude, C is a constant determined from the universal turbulence spectra and $\sigma$ is the standard deviation of the horizontal wind.

This dataset is described in Roberts et al. (2014) and compared to station observations. They found that modelled and observed wind gust speeds were generally in close agreement. However, less agreement was found when considering stations





with altitude greater than $\sim 500$ m. This is a common issue in atmospheric models (Donat et al., 2010), caused by the use of an effective roughness parameterisation, needed to estimate the effect of sub-grid scale orography on the synoptic scale flow, causing unrealistically slow wind speeds at 10 metres (Roberts et al., 2014). In addition, the model slightly underestimates extreme wind gust speeds greater than $\sim 25 \mathrm{ms}^{-1}$. This was found to be due to several mechanisms, for example, the underestimation of convective effects and strong pressure gradients, likely due to the limitations of the model horizontal resolution.

## 3 Identifying a damaging footprint characteristic

European windstorm footprints have commonly been used to represent the extremity of a windstorm event, either in the current climate, (e.g., Klawa and Ulbrich 2003, Haylock 2011, Bonazzi et al. 2012, Cusack 2013, Roberts et al. 2014), or under future climate change conditions, (e.g., Leckebusch et al. 2007, Pinto et al. 2007, Leckebusch et al. 2008, Donat et al. 2011, Pinto et al. 2012). In each study the severity of an event is measured using a Storm Severity Index (SSI), i.e. a conceptual loss function.

Klawa and Ulbrich (2003) developed an SSI for the estimation of windstorm losses in Germany. For windstorm event $i$, this SSI, here denoted $L_{98}$, was defined as

$$L_{98i} = \sum_{j=1}^{J} d(s_j) \left( \frac{v_i(s_j)}{v_{98}(s_j)} - 1 \right)^3 \quad \text{for } v_i(s_j) > v_{98}(s_j), \tag{1}$$

where $v_i(s_j)$ is the footprint wind gust speed at location $s_j$, $j = 1, ..., J$, for windstorm event $i$, $i = 1, ..., n$, $v_{98}(s_j)$ is the damage threshold for location $s_j$, the $98^{\text{th}}$ percentile of the climatology wind gust speed over the period of interest, and $d(s_j)$

is the population density at location $s_j$. The SSI was calculated for locations over land only to represent where insured property losses occur.

This SSI is based on a cubic relationship between wind gust speed and loss. Klawa and Ulbrich (2003) argued that, from a theoretical point of view, the cube of the wind speed is proportional to the damaging power, i.e. the rate of kinetic energy

delivered by advection. Empirically this relationship was supported by MunichRe (1993), who found that the loss extent of windstorms increased with almost the cube of the maximum wind gust speeds. Prahl et al. (2015), however, argued that the subtraction of the damage threshold in the Klawa and Ulbrich (2003) SSI resulted in an inconsistency in the cubic dependence on wind gust speed, and found that the gradient of this SSI was, in fact, much steeper than cubic. Klawa and Ulbrich (2003) noted that insurance companies in Germany pay for storms when maximum gusts are above $20 \mathrm{\ ms}^{-1}$ and consequently found

that this value coincided with the $98^{\text{th}}$ percentile of daily maximum gust wind speed in German flatland stations. They argued that both buildings and nature adapt to the local wind conditions and, as a result, the damage threshold should vary throughout the domain. In addition, Klawa and Ulbrich (2003) included a parameter for population density as a proxy for insured exposure in areas affected by damaging winds. The area of damaging winds is also represented in this loss function since the SSI is a sum over all locations experiencing damaging winds.






In subsequent studies such as Leckebusch et al. (2007), Pinto et al. (2007), Donat et al. (2010), Donat et al. (2011), and Pinto et al. (2012), SSIs had similar composition to Klawa and Ulbrich (2003). In addition, Cusack (2013) used a variation on the Klawa and Ulbrich (2003) SSI, based on the cube root of $L_{98}$, divided by the number of location in which damaging winds occur. Since the Klawa and Ulbrich (2003) SSI has been so widely used, it has been rigorously validated for its suitability for representing windstorm loss and damage. General agreement was found between the SSI and both annual aggregate losses and individual event losses, even for low resolution climate model data. However, validation was based solely on Germany due to

the unavailability of insured loss data for other countries.

Roberts et al. (2014) explored how well a number of SSIs represented insured loss throughout Europe when developing a method for selecting storms for the eXtreme Wind Storms (XWS) catalogue. They found that the SSI characterising the area of the footprint exceeding 25 ms$^{-1}$ over land, outperformed the SSI developed by Klawa and Ulbrich (2003) when classifying

a set of 23 extreme insurance loss windstorms. This damage area SSI was found to be the best footprint based index at representing insured loss. The most successful classifier was created by combining the damage area SSI with the cubed maximum 925hPa wind speed along the storm track, creating an SSI similar to that used by Lamb and Frydendahl (1991).

Following the conclusions of Roberts et al. (2014), a damage area SSI will be used here to represent the damage incurred

by a windstorm event. A damage threshold of 20ms$^{-1}$ will be used. This is slightly lower than the 25ms$^{-1}$ threshold used by Roberts et al. (2014), in order to reduce the sampling uncertainty in locations where the wind gust speeds are weaker and therefore rarely exceed 25ms$^{-1}$. This threshold choice is consistent with the damage threshold used by Bonazzi et al. (2012) and the damage threshold used by insurance companies in Germany, as identified by Klawa and Ulbrich (2003). The SSI for windstorm event $i$ is defined as:

$$A_{20i} = \sum_{j=1}^{J} H(v_i(s_j) - 20) \qquad (2)$$

where $H(x)$ is the Heaviside step function, $H(x) = 1$ if $x > 0$ and $H(x) = 0$ otherwise. As in Klawa and Ulbrich (2003) and Roberts et al. (2014), this SSI is calculated over land locations only.

Figure 2 shows a scatter plot of the logarithm of the SSI developed by Klawa and Ulbrich (2003), labelled $L_{98}$ (Eqn. 1)

and the damage area SSI $A_{20}$ (Eqn. 2), for the top 50% of the 6103 windstorm events in the data set in each SSI. Both SSIs are calculated using the $J = 14872$ grid cells over land. The local 98$^{\text{th}}$ percentile damage threshold in $L_{98}$ is calculated from daily maximum wind gust speeds in the period October 1979 - March 2012. The population density data in $L_{98}$ is taken from Haylock (2011), created based on the 2005 Gridded Population of the World version 3, re-gridded to the spatial resolution of the footprint data. Figure 2 shows that there is a positive association, indicating that $A_{20}$ can be used to predict the more

complex SSI, $L_{98}$. In addition the 23 notable, damaging windstorms used by Roberts et al. (2014) to explored how well SSIs represented insured loss, are shown in Figure 2. The 23 extreme loss storms occur for extremes in both SSIs, although are better



classified by $A_{20}$, i.e. when $\log(A_{20} > 7.3)$. Storms that cause large insured losses therefore generally have a large footprint area of wind gust speeds exceeding 20 ms$^{-1}$.

## 4 How has this damaging footprint characteristic changed?

Figure 3 shows the October - March winter variability in the number of windstorm events, the winter average value of $A_{20}$ and the winter total $A_{20}$. The number of events remains relatively constant throughout the 20$^{\text{th}}$ century with very little variation
between years. The number of events shows an increase in the 21$^{\text{st}}$ century, with much greater variation between years. However, winter averaged and total $A_{20}$ has declined in the 21$^{\text{st}}$ century. This suggests, in agreement with Cusack (2013), that the number of very damaging windstorms has decreased in recent decades.

The large spatial domain covered by the available footprint data allows for the spatial variation in this change in $A_{20}$ to be
explored. The relative frequency of exceeding 20 ms$^{-1}$ is calculated for each location as

$$F(s_j) = \frac{1}{n} \sum_{i=1}^{n} H(v_i(s_j) - 20)$$

where $v_i(s_j)$ is the footprint wind gust speed at location $s_j$, $j = 1, ..., J$, for windstorm event $i$, $i = 1, ..., n$ and $H(x)$ is the Heaviside step function, $H(x) = 1$ if $x > 0$ and $H(x) = 0$ otherwise. This relative frequency is calculated separately at each location for windstorm events in winters associated with the 20$^{\text{th}}$ century (1979/80 - 1999/00) and the 21$^{\text{st}}$ century (2000/01
- 2013/14), denoted $F(s_j)$ and $F'(s_j)$ respectively, to explore how this damage threshold exceedance has changed in the two periods at different locations throughout the European domain. The number of events in each period is $n = 3654$ and $n' = 2554$ respectively.

Figure 4 (a) shows the relative frequency of exceeding 20 ms$^{-1}$ at each location in the earlier period ($F(s_j)$) and Fig. 4
(b) shows the ratio of relative frequencies in the later and earlier periods ($F'(s_j)/F(s_j)$). The relative frequency has decreased in a majority of locations throughout Europe, particularly in the north of the continent. An increase has occurred in more southerly locations such as Spain and the eastern Mediterranean, suggesting that wind gust speeds have, in fact strengthened in southern Europe in the later period. However, $F(s_j)$, is very low in these regions (0.005-0.06), therefore this increase is relatively small compared to the decrease in northern Europe.

The statistical significance can be tested using a two-sample t-test for equal means. The following statistic is calculated at each of the land grid points:

$$t(s_j) = \frac{F'(s_j) - F(s_j)}{\tilde{s}\sqrt{1/n + 1/n'}}, \tag{3}$$





where $\tilde{s}^2$ is the pooled variance

$$\tilde{s}^2 = \frac{(n-1)\tilde{s}_1^2 + (n'-1)\tilde{s}_2^2}{n+n'-2} \tag{4}$$

and $\tilde{s}_1$ and $\tilde{s}_2$ are the standard deviation of the binary exceedance variable $H(v-20)$ in the earlier and later periods respectively. The change in relative frequency is statistically significant at level $\alpha$ if $|t| > t_{1-\alpha/2,\nu}$, where $\nu = n + n' - 2$.

Figure 4 (c) shows the test statistic, $t$, calculated for each land grid cell, where only those locations that experience a statistically significant change in relative frequency at a $5\%$ level are shown in colour. A significant decrease is identified in the UK, northern France, Germany, Belgium, the Netherlands and Denmark, as well as along the coast of north-eastern Europe and in western Russia. A significant increase is identified in eastern and western Spain and in the small region of northern Africa included in the European domain. Southern France, Norway, Sweden and most of eastern Europe have seen no significant change. The decline in the footprint damage area $A_{20}$ in the the $21^{\text{st}}$ is therefore primarily due to a decrease in the frequency of winds exceeding $20\ \text{ms}^{-1}$ in north-west Europe.

The Quantile-Quantile plot of the sample of footprint wind gust speeds in the grid cell closest to Paris (white cross in Fig. 4 (c)) for events in the two comparative periods is displayed in Fig. 5. This location is chosen due to its large, significant decrease in relative frequency. The quantiles are approximately equal for wind gust speeds below $18\ \text{ms}^{-1}$. Above this threshold the quantiles for the later period are lower than those for the earlier period, for example the extremity of wind gust speeds of $20\ \text{ms}^{-1}$ in the earlier period corresponds to the extremity of approximately $19\ \text{ms}^{-1}$ in the later period. A similar relationship between the two periods is found at locations throughout north-west Europe.

## 5 Why has this damaging footprint characteristic changed?

The NAO is the dominant mode of lower to mid-tropospheric pressure variability over the North Atlantic (Pinto et al., 2012). For high values of the NAO index, which often occurs in winter, pressure differences between the Icelandic Low and Azores High increase, hence increasing the frequency of low-pressure systems, leading to increased storm genesis. In the opposite NAO phase, a low pressure difference leads to below average winds and the southward displacement of low-pressure systems.

The relationship between NAO and severe windstorms has been explored in a number of studies. For example Raible (2007) identified an NAO+ pattern in the sea-level pressure field when correlated with the occurrence of extreme cyclones in Northern Europe. Matulla et al. (2008), however, found that the ability for the NAO index to explain storminess in Europe varied in space and depended on the period analysed. More recently, Economou et al. (2014) developed a spatial-temporal statistical model for sea-level pressure. They used the NAO index as a covariate in the model and found it to have a significant effect on intensifying extremal storm behaviour, especially in Northern Europe and the Iberian peninsula.





Figure 6 shows the correlation between the winter total $A_{20}$ and the winter averaged mean sea-level pressure in each grid cell in the global ERA-interim dataset (Dee et al., 2011) for winters (October - March) 1979/80 - 2013/14. The correlation pattern resembles the NAO pattern, albeit shifted somewhat eastwards over Europe. This suggests that windstorms with large values of $A_{20}$ may be associated with positive NAO and hence that much of the change in $A_{20}$, and therefore windstorm losses, can be explained by NAO. The eastward shift in the correlation pattern, however, suggests that NAO is not the only driver in the change in windstorm losses.

This role of NAO is confirmed by Fig. 7 which shows the strong positive association between winter total $A_{20}$ and winter

averaged NCEP CPC NAO index (NCEP, 2016), a correlation of 0.715. In addition, the linear fit between these variables (solid line in Fig. 7), identifies that winter averaged NAO accounts for 51% of the total variance in winter total $A_{20}$ and has a slope that is significantly non-zero (p-value $1.38 \times 10^{-6}$). The two comparative periods, winters in the 20th and 21st centuries, show different characteristics, with the earlier period generally having higher values of NAO and $A_{20}$ compared to the later period. However, the winters of 1989/90 and 2013/14, which had very different windstorm losses, as discussed in the Introduction,

have a very similar NAO index. This further suggest that factors other than NAO are influencing windstorm losses.

## 6   Conclusions

The 21st century decline in damaging windstorms and related insured losses is investigated by identifying a windstorm footprint characteristic that represents wind related damage and exploring how and why this characteristic has changed in recent decades.

The area of the footprint exceeding 20 ms$^{-1}$ over land is shown to be a good predictor of windstorm damage. This SSI, denoted $A_{20}$, has a strong positive relationship with the SSI developed and validated by Klawa and Ulbrich (2003), but was found to be more representative of extreme insurance loss windstorms.

The October - March winter average and total $A_{20}$ was shown to have decreased in the 21st century, mirroring the recent

decline in windstorm related insured losses. The number of windstorm events in each winter was found to have increased in the same period, suggesting that the decline in windstorm damage is due to a decrease in the number of very damaging windstorms. This decline in $A_{20}$ between 20th and 21st centuries was found to be due to a significant decrease in the relative frequency of exceeding 20ms$^{-1}$ in north-west Europe. This decrease was shown to be because of a decrease in the quantiles of the footprint wind gust speed distribution above approximately 18 ms$^{-1}$ at these locations.


The correlation pattern between winter total $A_{20}$ and winter averaged mean sea level pressure was shown to resemble the NAO pattern, shifted eastwards over Europe. A strong positive association was also found between winter total $A_{20}$ and winter averaged NAO, suggesting that much of the change in $A_{20}$ is explained by NAO. In addition, the winter averaged NAO is found to account for 51% of the winter-to-winter variation in total $A_{20}$. The eastward shifted NAO pattern and the very similar NAO



indices in the winters of 1989/90 and 2013/14, however, suggest that other modes of variability play a role in the variation in windstorm losses.

In future research it would be desirable to further explore which factors, other than NAO, have an influence on $A_{20}$. This will give a better understanding of the reason for this recent decline in windstorm losses, particularly in the winter of 2013/14. In addition, the development of a statistical model of windstorm footprints would allow for a better understanding of how statistical properties of windstorm footprints, for example the spatial dependence between locations, have changed causing this decline in windstorm losses. This has been addressed in Dawkins (2016), soon to be written as a publication.

5  *Acknowledgements.* We would like to acknowledge Mathias Graf for constructive discussions about losses from windstorm footprints. Laura C. Dawkins was supported by the Natural Environment Research Council (Consortium on Risk in the Environment: Diagnostics, Integration, Benchmarking, Learning and Elicitation (CREDIBLE project); NE/J017043/1).



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





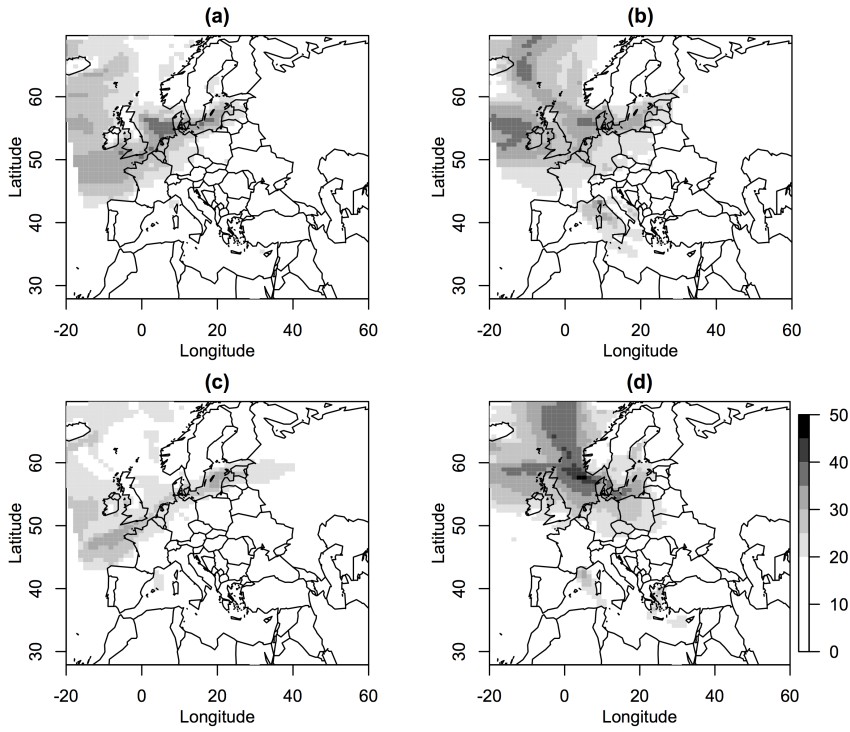

**Figure 1.** Windstorm footprints for (a) Daria (24[th]-26[th] January 1990), (b) Vivian (25[th]-27[th] February 1990), (c) Christian (27[th]-29[th] October 2013) and (d) Xaver (4[th]-6[th] December 2013). The footprint is defined as the maximum 3 second wind gust speed (in ms$^{-1}$) over a 72 hour period centred on the time at which the maximum 925hPa wind speed occurs over land, estimated by dynamically downscaling ERA-Interim reanalysis to a 25 km horizontal resolution using the Met Office unified model (see Section 2).



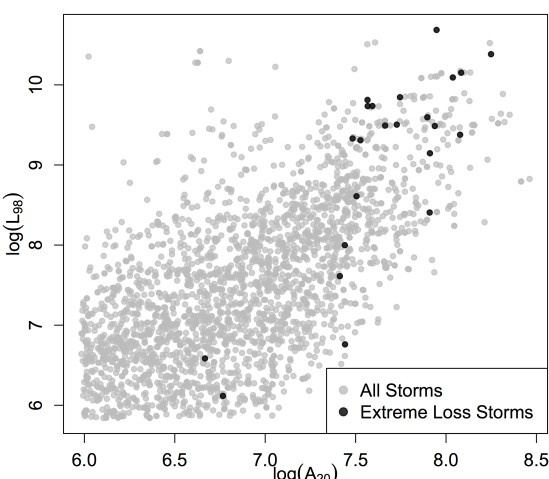

**Figure 2.** Relationship between the logarithm of the Klawa and Ulbrich (2003) loss function ($L_{98}$) and the logarithm of the damage area SSI ($A_{20}$) for the top 50% of the 6103 windstorms in each variable. The 23 notable, large insurance loss storms, defined in Roberts et al. (2014), are shown in black.





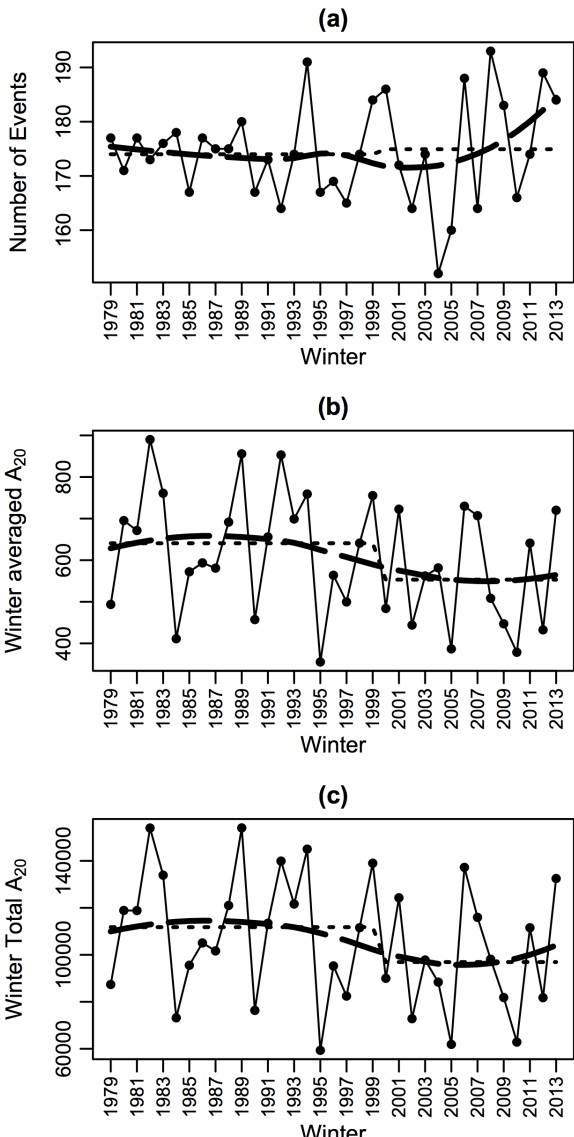

**Figure 3.** Winter (October - March) variability in (a) the number of windstorm events, (b) average $A_{20}$ and (c) total $A_{20}$. Trends are depicted by locally weighted scatterplot smoothing curves (dashed curves). Dotted lines show means in each variable in the 20[th] century (1979/80-1999/00) and the 21[st] century (2000/01-2013/14).




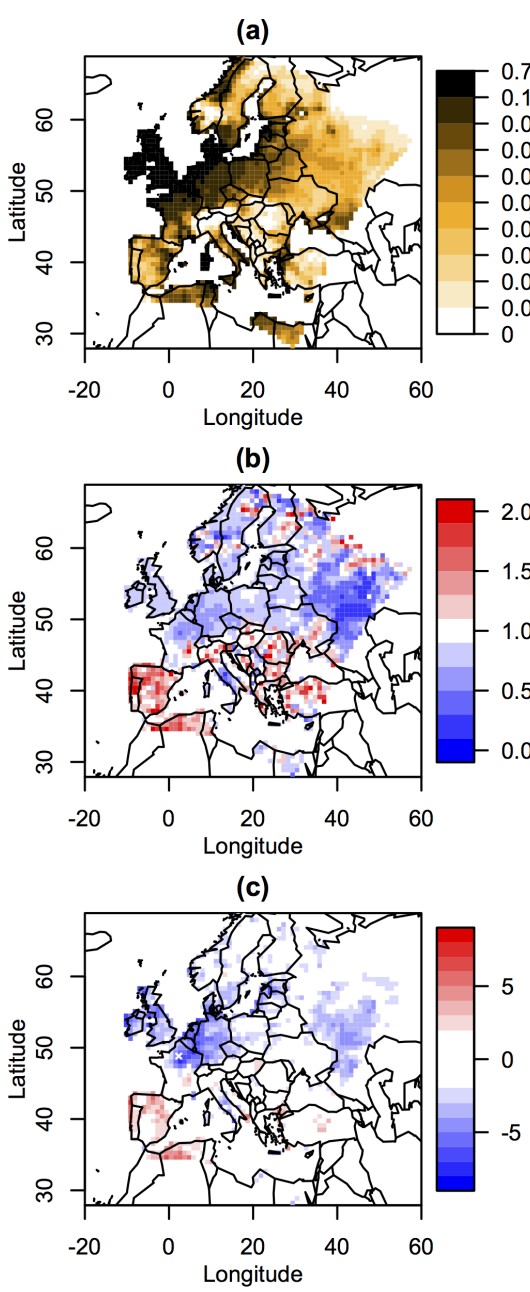

**Figure 4.** (a) Relative frequency of exceeding 20 ms$^{-1}$ in the 20$^{th}$ century, $F(s_j)$, (b) ratio of the relative frequencies in the 21$^{st}$ and 20$^{th}$ centuries, $(F'(s_j)/F(s_j))$, and (c) the test statistic, $t$, for the two-sample t-test for equal means for the relative frequencies. The grid cell closest to Paris is indicated by a white cross, the location explored in Figure 5.



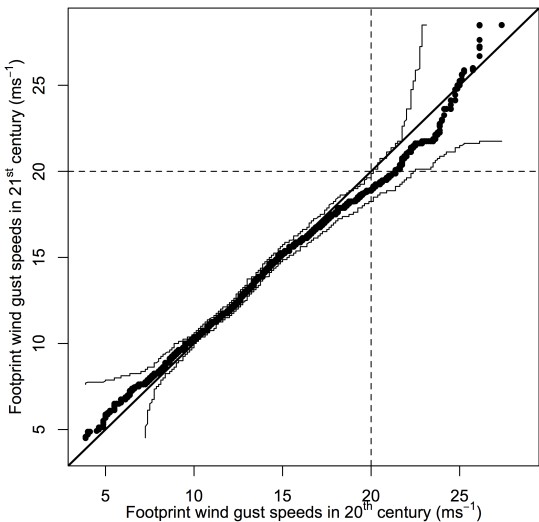

**Figure 5.** Quantile-Quantile plot of footprint wind gust speeds in the grid cell closest to Paris for events in the two comparative periods: winters in the 20[th] century (1979/80 - 1999/00) and winters in the 21[st] century (2000/01 - 2013/14). The thick solid black line shows where y=x, the dashed black lines show the 20 ms$^{-1}$ damage threshold and the thin solid black lines show the 95% confidence interval, based on the asymptotic sampling distribution of the order statistics.

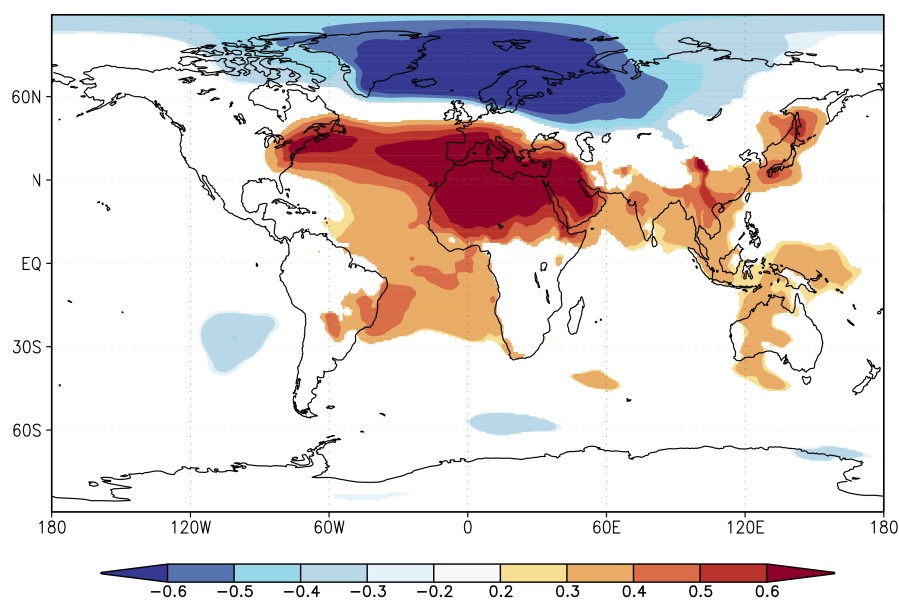

**Figure 6.** Correlation between winter total $A_{20}$ and winter averaged mean sea-level pressure in each grid cell in the ERA-interim reanalysis for October - March winters 1979/80 - 2013/14.





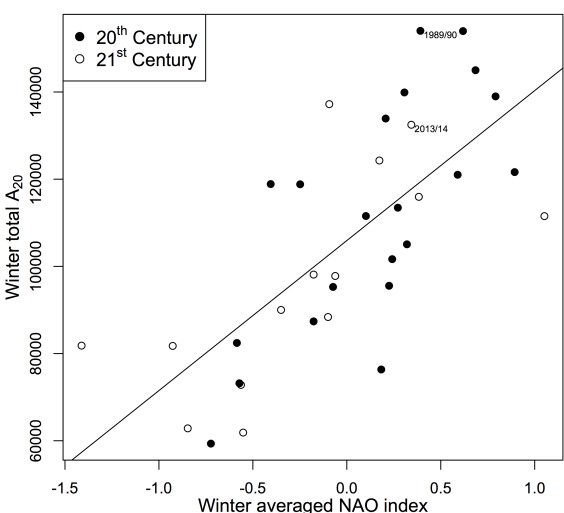

**Figure 7.** Relationship between winter total $A_{20}$ and winter averaged NAO index for October - March winters 1979/80 - 2013/14. The linear fit (y∼x) is shown (solid line). The two comparative periods, winters in the 20th century and winters in the 21st century, are indicated. The points associated with winters 1989/90 and 2013/14, discussed in the Introduction, are labelled.