# Peer review of "The 21st Century Decline in Damaging European Windstorms"

_Natural Hazards and Earth System Sciences, 2016_

## Referee Comment (RC1) · Anonymous Referee #1 · 9 May 2016

General Comments

The authors use spatial footprints of gusts from the XWS catalogue of extra-tropical cyclone events in the past 35 years to examine the decline in windstorms over parts of Europe in the past 15 years. The authors are to be commended for a very clear writing style. I recommend publication after some further analysis and revisions.

The main comment is that there is insufficient evidence that the A20 metric is valid for its purpose of documenting a recent decline in damaging European windstorms. The validation in the article consists of finding a value of X% containing 23 significant windstorms, and a lower value of X indicates a better metric. This metric analyses a subjective subset of individual severe windstorms, and is not robust to outliers. Further, the validation concerned an A25 rather than the A20 metric used in this article. However, there is a much bigger validation problem: this article analyses annual integrated A20 rather than individual severe windstorms, hence the proposed validation has little relevance. The distinction between individual event damage and annual integrated values is very substantial here: we know that a single storm such as Daria or Lothar produced more damage than the long-term annual average, whereas the A20 estimate for Daria is 1 or 2% of long-term annual average and points to far too much weight on weak storms in A20. The authors have to choose a metric which has been validated as an annual integrated measure of damage. Articles such as Barredo (2010) can provide some data on annual integrated damage to help with such a validation. This is not viewed as a major change in direction since the authors use metrics such as Klawa and Ulbrich (2003) in this article - instead, it is a change in emphasis on the best metric.

In general, the article follows the paradigm 'large-scale climate patterns force weather events' with statements similar to 'NAO explains changes in A20'. However, the aggregate of the individual weather events contribute significantly to the large-scale pattern, see text and references within "Contrasting interannual and multidecadal NAO variability" by Woollings et al. (2014). Would the authors consider changing their description of climate-weather link to something like 'changes in NAO are consistent with annual damage metric variations', if they find such behaviour?

Specific Comments

Page 1, lines 18-22: Roberts et al. do not provide loss estimates for the four named storms, could the source of these losses be given?

Page 2, line 11: the loss function used five stations, rather than four.

Page 2, lines 17-19: the published work by Wallace would be a better reference for the NAO?

Page 6, lines 3-7: The statement "..number of very damaging windstorms has decreased in recent decades" is not supported by the evidence in Figure 3. Instead,

the decline in total A20 is caused by reduced A20 per event (Fig 3b) and there is no evidence specific to the subset of very damaging storms in Figure 3.

Page 7, lines 8-9: this sentence is tautological: A20 counts number of occurrences of wind > 20 m/s, or put another way, is a measure of frequency of occurrence of winds > 20 m/s.

Page 7, lines 11-16: Figure 5 helps to explain A20 changes in Paris, but it also raises some major issues. First, the peak gusts never exceed 30 m/s, yet storm Lothar was measured above 40 m/s by multiple weather stations in and around Paris. Second, the top 10 or 20 storms in the period are responsible for the vast majority of damage at Paris, and the top 10 points in this plot show the recent period to have consistently higher gusts. This indicates the extreme gusts from XWS footprints are very different from observed behaviour. Do the authors if XWS wind values have been compared to actual weather station gusts, and if so, has a trend been found such that modelled hazard for older storms have more negative bias with respect to observed, compared to newer storms?

Page 8, lines 4-10: could the Scandinavia Pattern be included in analysis? The eastwards extent of the spatial pattern in Figure 6 suggests the SP.

Page 8, lines 20-21: the conclusion to be drawn from lower total A20 and higher number of events is the mean A20 per event is lower. No conclusion can be drawn about very damaging windstorms, since they are a very small part of this particular A20 metric (see comments about Paris above, where the q-q plot indicates more severe storms in past 15 years).

Page 8, lines 22-24: these two sentences are redundant.

---

## Referee Comment (RC2) · Dr. Graf (Referee) · 13 May 2016

General comments:

The excellent work of the author has definitely a quite real impact on the insurance industry. This study helps to understand and explain the observed discrepancy between the (in the insurance industry commonly used) windstorm risk models (build on 40 years of meteorological data) and the real claims data (captured over the last 10 years). Which helps to derive a more realistic risk view for the coming years.

The paper is very clearly, consistently and illustratively write and I definitely recommend publication.

Detailed comments:

[Figure]

The following comment is rather a suggestion how the quality could be improved e.g. in a follow on study. The used 25km resolution of the windstorm footprints is rather a low resolution. For example in tropical cyclone risk models normally a resolution of around 1km is used to best estimate the damage. Several studies in insurance companies (unfortunately unpublished to my knowledge) show an increase of the correlation between claims data and reproduced wind speeds with a higher resolution, even if a statistical down scaling is used. Common practice is to use a surface friction model as described for example in "Meng, Y., M. Matsui and K. Hibi (1997) A numerical study of the wind field in a typhoon boundary layer, Journal of Wind Engineering and Industrial Aerodynamics, 67&68, pp. 437-448." and estimating the surface friction based on land use data as for example described in "Graf, M., K. Nishijima and M. H. Faber (2009) A Probabilistic Typhoon Model for the Northwest Pacific Region, APCWE 7, Taipei.". This would help to derive more realistic local maximum wind speeds. A higher resolution could help to derive more realistic wind speeds compared to shown (low) wind speeds in the Paris example.

But since the author is investigating an explanation of a "relative" decline of damaging European windstorms, rather than an absolute value, I strongly assume that the proposed suggestion will not have any impact on the final conclusion.

The findings on page 6, describing the increase in the variability of the storm activity in the recent years and describing the increase of the frequency of exceeding 20 m/s wind speeds in southern Europe, are also very interesting discoveries and should be mentioned in the conclusions and/or in the abstract.

As mentioned already in the outlook, it definitely would be interesting to investigate if there are more driving factors aside of NOA which influence the storm size.

Figure 1: The legend of the color bar is missing, I assume it should be max 3 sec gust wind speed [m/s] as stated in the figure description.

Page 5: Adding the value of the correlation coefficient between log(L98) and log(A20)

[Figure]

could help to describe the positive correlation, e.g. in Figure 2.

Figure 5: It would may be worthwhile to set the limits for the x and y axis to 10 – 30 m/s, since the increase in the uncertainty below 10m/s is probably just related to the number of storms for which a footprint was generated and this would magnify the tail of the distribution which is relevant for the argumentation.

[Figure]

---

## Author Comment (AC1) · 14 Jul 2016

General comments

Referee comment:

The main comment is that there is insufficient evidence that the A20 metric is valid for its purpose of documenting a recent decline in damaging European windstorms. The validation in the article consists of finding a value of X% containing 23 significant windstorms, and a lower value of X indicates a better metric. This metric analyses a subjective subset of individual severe windstorms, and is not robust to outliers. Further, the validation concerned an A25 rather than the A20 metric used in this article.

Response:

This is a very good point which should be addressed within the text. This validation method was used in the Roberts et al (2012) paper and remains the best method here, because the insured loss value for the whole of Europe is only available to us for a very limited number of events, all of which are extreme storms (see Table 1 in Roberts et al. (2012)), not enough to validate the SSI. Even the Barredo (2010) paper only includes losses from 54 extreme storms between 1970 and 2008, still thought to be too few to validate these SSIs. Since the A25 measure was validated in the Roberts et al (2012) paper and has a strong positive relationship with A20 (see attached scatter plot), we feel this is sufficient evidence that the A20 SSI is valid for its purpose in this paper, having no further way to validate it.

This issue will be addressed by adding a paragraph on page 5, line 13:

'The SSI validation method used by Roberts et al. (2012) is based on a subjective subset of extreme windstorms and is not robust to outliers. However, since the value of insured loss for the whole of Europe is only available for a very limited number of windstorm events, all of which are extreme (Roberts et al., 2012 (Table 1); Barredo, 2010), this is thought to be the most appropriate available method for validation here. If further loss data were made available for validation the exploration of the most successful European-wide SSI would be a hugely beneficial area of future research.'

We will amend Figure 2 to include the scatter plots of A25 and A20 and add text on page 5, line 24:

'Figure 2 (a) shows a scatter plot of the logarithm of the damage area SSIs A25 and A20, and Figure 2 (b), a scatter plot of the logarithm of the SSI developed by Klawa and Ulbrich (2003), labelled L98 and the damage area SSI A20, for the top 50% of the 6103 windstorm events in the data set in each SSI.'

Define A25 on page 5, line 8/9:

'They found that the SSI characterising the area of the footprint exceeding 25 ms$-1$

over land, A25, outperformed ...'

And page 5, line 29:

'Figure 2 (a) shows that there is a strong positive relationship between A20 and A25, indicating that the A20 SSI is appropriate for representing insured loss, as shown by Roberts et al. (2012) for the A25 SSI.'

Referee comment:

However, there is a much bigger validation problem: this article analyses annual integrated A20 rather than individual severe windstorms, hence the proposed validation has little relevance. The distinction between individual event damage and annual integrated values is very substantial here: we know that a single storm such as Daria or Lothar produced more damage than the long-term annual average, whereas the A20 estimate for Daria is 1 or 2% of long-term annual average and points to far too much weight on weak storms in A20. The authors have to choose a metric which has been validated as an annual integrated measure of damage. Articles such as Barredo (2010) can provide some data on annual integrated damage to help with such a validation. This is not viewed as a major change in direction since the authors use metrics such as Klawa and Ulbrich (2003) in this article - instead, it is a change in emphasis on the best metric.

Response:

We acknowledge the difference between A20 for individual severe storms and annual integrated A20, however we are of the view that an SSI that is able to rank extreme storms highly and therefore weaker storms lower, can represent the insured loss in general. In addition, since we have the footprints for all windstorm events in each year, the sum of A20 over all of these events in one year, should be equivalent to the annual aggregate loss.

We are unsure of where these figures (1 or 2%) have come from. For windstorm Daria,

[Figure]

A20 is approximately 3000 (Figure 2 is on a logarithmic scale exp(8)∼3000), which is about 500% of the long-term annual average A20 (∼600). Has the Figure been misread or can this comment be explained further please?

As mentioned above, the Barredo (2010) paper contains losses for 54 extreme events between 1970 and 2008, therefore we are of the view that the annual aggregate losses used in this paper will not be comparable to the annual aggregate A20 values.

Additional mention of the caveats of the A20 SSI will be included in the conclusion-Page 8, line 17:

'This conclusion is based on a subjective set of extreme windstorms and could therefore benefit from further validation based on insured loss data, if more were available. In addition, other SSIs could have been included in the validation investigation, using different damage thresholds and exposure variables. Many such SSIs are compared by Dawkins (2016), where the damage area SSI A25 is still shown to be most successful at representing insured loss. Further, this validation is based on relatively low resolution footprints and could give different conclusions if higher resolution footprints were available for exploration.'

Referee comment:

In general, the article follows the paradigm 'large-scale climate patterns force weather events' with statements similar to 'NAO explains changes in A20'. However, the aggregate of the individual weather events contribute significantly to the large-scale pattern, see text and references within "Contrasting interannual and multidecadal NAO variability" by Woollings et al. (2014). Would the authors consider changing their description of climate-weather link to something like 'changes in NAO are consistent with annual damage metric variations', if they find such behaviour?

Response:

This is a very good point. The language is too causal in some places and will be

changed.

Page 8, line 32:

'This suggests that windstorms with large values of A20 may be associated with positive NAO and hence that much of the variation in A20, and therefore windstorm losses, *is consistent with variation in NAO*. '

Page 8, line 27:

'A strong positive association was also found between winter total A20 and winter averaged NAO, showing that changes in NAO are consistent with the variation in annual A20 and is therefore related to annual loss.'

Specific Comments

Referee comment:

Page 1, lines 18-22: Roberts et al. do not provide loss estimates for the four named storms, could the source of these losses be given?

Response:

Table 1 in Roberts et al. (2012) gives the losses estimates for 16 extreme windstorms including these four named storms. These loss estimates are sourced from the Sigma technical reports between 2004 and 2013.

Referee comment:

Page 2, line 11: the loss function used five stations, rather than four.

Response:

Thank you, this will be amended

Referee comment:

Page 2, lines 17-19: the published work by Wallace would be a better reference for the

NAO?

Response:

This is a good point. On reconsideration of this reference, we think the most appropriate reference is the 2003 book by James W. Hurrell et al:

Hurrell, J. W., Y. Kushnir, G. Ottersen and M. Visbeck, 2003: The North Atlantic Oscillation, Climate Significance and Environmental Impact. American Geophysical Union, Washington, DC.

Referee comment:

Page 6, lines 3-7: The statement "..number of very damaging windstorms has decreased in recent decades" is not supported by the evidence in Figure 3. Instead, the decline in total A20 is caused by reduced A20 per event (Fig 3b) and there is no evidence specific to the subset of very damaging storms in Figure 3.

Response:

Thank you, this is a very good point. This statement will be removed from the text since this conclusion cannot be taken from Figure 3.

Referee comment:

Page 7, lines 8-9: this sentence is tautological: A20 counts number of occurrences of wind > 20 m/s, or put another way, is a measure of frequency of occurrence of winds > 20 m/s.

Response:

This sentence will be reworded to bring out the emphasis on north-west Europe: 'The largest contribution to the decline in A20 comes from north-west Europe.'

Referee comment:

Page 7, lines 11-16: Figure 5 helps to explain A20 changes in Paris, but it also raises

some major issues. First, the peak gusts never exceed 30 m/s, yet storm Lothar was measured above 40 m/s by multiple weather stations in and around Paris. Second, the top 10 or 20 storms in the period are responsible for the vast majority of damage at Paris, and the top 10 points in this plot show the recent period to have consistently higher gusts. This indicates the extreme gusts from XWS footprints are very different from observed behaviour. Do the authors if XWS wind values have been compared to actual weather station gusts, and if so, has a trend been found such that modelled hazard for older storms have more negative bias with respect to observed, compared to newer storms?

Response:

These are very interesting points. Weather stations, which are uncertain themselves, provide point observations of wind speeds, whereas what we present is a gridded analysis so we would not expect them to be the same. However, it is true that this bias can be large. The XWS gridded footprints used here have been shown to underestimate wind gust speeds above 25m/s – see Figure 8 f) in Roberts at al. (2012), where an observed wind gust of ∼60m/s is modelled as being ∼20m/s. This bias is likely due to the low resolution of the gridded analysis.

When developing the XWS catalogue, only the 50 storms included in the catalogue were recalibrated, and therefore compared to weather station data. No temporal trend in the bias was noticed, however, this could be interesting to explore in more detail in future work.

We will address these comments by including extra text in data section – page 4, line 3:

'In addition, the gridded analysis is shown to underestimate extreme wind gust speeds greater than âĹij 25ms−1 to varying degrees. Figure 8 c) and f) in Roberts et al (2012) show how this underestimation is small for some locations but in others this bias is much larger, with observed wind gust speeds greater than 40ms-1, modelled as being

below 30ms-1. This bias was found to be due to several mechanisms, for example, the underestimation of convective effects and strong pressure gradients, likely due to the limitations of the model horizontal resolution.'

By adding text after the q-q plot – page 7, line 15:

'The most extreme wind gust speeds are higher for the 21st century, however, the bias in modelled wind gust speeds at this high level, shown by Roberts et al (2012) and discussed in Section 2, suggest that these high quantiles should be treated with great uncertainty. For example, it is known that during windstorm Lothar in December 1999 wind gust speeds near 50ms-1 were recorded in Paris (Ulbrich et al 2001).'

Additional reference: Ulbrich, U., A. H. Fink, M. Klawa and J. G. Pinto, 2001: Three extreme storms over Europe in December 1999. Weather, 56: 70-80. doi:Âă10.1002/j.1477-8696.2001.tb06540.x

And lastly, adding a couple of sentences to the conclusion – page 9, line 32:

'This investigation is based on model generated gridded analysis windstorm footprints. While the large number of footprints within the data set is beneficial is validating conclusions, the modelled wind gust speeds are known to be biased, particularly in area of high altitude and when wind gust speeds exceed 25ms-1. Further exploration of this bias could improve the validity of the investigation and carrying out the same analysis using both observations and gridded analysis could be an interesting extension of this work. '

Referee comment:

Page 8, lines 4-10: could the Scandinavia Pattern be included in analysis? The eastwards extent of the spatial pattern in Figure 6 suggests the SP.

Response:

We are of the view that the Scandinavian Pattern will not relate to correlation pattern

over both north and south Europe. In addition, we think that in adding more climate indices, a longer time series is required to identify relationships and we are restricted to 35 years of footprint data.

Referee comment:

Page 8, lines 20-21: the conclusion to be drawn from lower total A20 and higher number of events is the mean A20 per event is lower. No conclusion can be drawn about very damaging windstorms, since they are a very small part of this particular A20 metric (see comments about Paris above, where the q-q plot indicates more severe storms in past 15 years).

Response:

Again, this is a very good point. This statement will be removed from the text since this conclusion cannot be taken from Figure 3

However, this does not relate to the q-q plot because, here, extreme is defined as exceeding the 20m/s damage threshold, not the actual wind intensity.

Referee comment:

Page 8, lines 22-24: these two sentences are redundant.

Response:

Again, this statement will be reworded: 'The largest contribution to the decline in A20 between 20th and 21st centuries comes from north-west Europe.'

———————————————————

[Figure]

**Fig. 1.**

---

## Author Comment (AC2) · 14 Jul 2016

General comments

Referee comment:

The excellent work of the author has definitely a quite real impact on the insurance industry. This study helps to understand and explain the observed discrepancy between the (in the insurance industry commonly used) windstorm risk models (build on 40 years of meteorological data) and the real claims data (captured over the last 10 years). Which helps to derive a more realistic risk view for the coming years. The paper is very clearly, consistently and illustratively write and I definitely recommend publication.

Response:

Thank you!

Detailed comments

Referee comment:

The following comment is rather a suggestion how the quality could be improved e.g. in a follow on study. The used 25km resolution of the windstorm footprints is rather a low resolution. For example, in tropical cyclone risk models normally a resolution of around 1km is used to best estimate the damage. Several studies in insurance companies (unfortunately unpublished to my knowledge) show an increase of the correlation between claims data and reproduced wind speeds with a higher resolution, even if a statistical down scaling is used. Common practice is to use a surface friction model as described for example in "Meng, Y., M. Matsui and K. Hibi (1997) A numerical study of the wind field in a typhoon boundary layer, Journal of Wind Engineering and Industrial Aerodynamics, 67&68, pp. 437-448." and estimating the surface friction based on land use data as for example described in "Graf, M., K. Nishijima and M. H. Faber (2009) A Probabilistic Typhoon Model for the Northwest Pacific Region, APCWE 7, Taipei.". This would help to derive more realistic local maximum wind speeds. A higher resolution could help to derive more realistic wind speeds compared to shown (low) wind speeds in the Paris example. But since the author is investigating an explanation of a "relative" decline of damaging European windstorms, rather than an absolute value, I strongly assume that the proposed suggestion will not have any impact on the final conclusion.

Response:

Thank you for these suggestions. This point could be addressed by adding a sentence in the conclusion, page 9, line 32:

'In future work, when such data is available, this investigation could be improved by using windstorm footprints at a higher grid cell resolution which may provide more

realist local maximum wind gust speeds and therefore be more representative of the damage caused by an event.'

Referee comment:

The findings on page 6, describing the increase in the variability of the storm activity in the recent years and describing the increase of the frequency of exceeding 20 m/s wind speeds in southern Europe, are also very interesting discoveries and should be mentioned in the conclusions and/or in the abstract.

Response:

This is a good idea. We will alter the second paragraph of the abstract to:

'The footprint of a windstorm is defined as the maximum wind gust speed to occur at a set of spatial locations over the duration of the storm. The area of the footprint exceeding 20 ms$-1$ over land, A20, is shown to be a good predictor of windstorm damage. This damaging characteristic has decreased in the 21st century, due to a statistically significant decrease in the relative frequency of windstorms exceeding 20 ms$-1$ in north-west Europe, although an increase is observed in southern Europe. The decrease in north-west Europe is explained by a decrease in the quantiles of the footprint wind gust speed distribution above approximately 18 ms$-1$ at locations in this region. In addition, an increased variability on the number of windstorm events is observed in the 21st century.'

And alter the third paragraph of the conclusion to:

'The October - March winter average and total A20 was shown to have decreased in the 21st century, mirroring the recent decline in windstorm related insured losses. The number of windstorm events in each winter and the variation in the number of event was found to have increased in the same period. This decline in A20 between 20th and 21st centuries was found to be due to a significant decrease in the relative frequency of exceeding 20ms$-1$ in north-west Europe. A significant increase in the

relative frequency of exceeding 20ms-1 is identified in southern Europe, however, this change is small in relation to the decrease in the north-west. The decrease in north-west Europe was shown to be because of a decrease in the quantiles of the footprint wind gust speed distribution above approximately 18 ms$-1$ at these locations.'

Referee comment:

As mentioned already in the outlook, it definitely would be interesting to investigate if there are more driving factors aside of NOA which influence the storm size.

Response:

We think that in adding more climate indices, a longer time series is required to identify relationships and we are restricted to 35 years of footprint data.

This could be commented on in the conclusion – page 9, line 32:

'In future research it would be desirable to further explore which factors, other than NAO, have an influence on A20. However, since many climate indices have multi-decadal cycles, this exploration would benefit from a longer time series of wind gust speeds than the 35 years of footprint data available here.'

Referee comment:

Figure 1: The legend of the color bar is missing, I assume it should be max 3 sec gust wind speed [m/s] as stated in the figure description.

Response:

Good spot, we will add this to the figure, thank you.

Page 5: Adding the value of the correlation coefficient between log(L98) and log(A20) could help to describe the positive correlation, e.g. in Figure 2.

Response:

Good idea, thank you.

Figure 5: It would may be worthwhile to set the limits for the x and y axis to 10 − 30 m/s, since the increase in the uncertainty below 10m/s is probably just related to the number of storms for which a footprint was generated and this would magnify the tail of the distribution which is relevant for the argumentation.

Response:

Good idea, we will do this.